# PUSHING THE LIMITS OF UNCONSTRAINED MACHINE-LEARNED INTERATOMIC POTENTIALS

## ABSTRACT

Machine-learned interatomic potentials (MLIPs) are increasingly used to replace computationally demanding electronic-structure calculations to model matter at the atomic scale. The most commonly used model architectures are constrained to fulfill exactly a number of physical laws, from geometric symmetries to energy conservation. Evidence is mounting that relaxing some of these constraints can be beneficial to the efficiency and (somewhat surprisingly) accuracy of MLIPs, even though care should be taken to avoid qualitative failures associated with the breaking of physical symmetries. Given the irresistible trend of *scaling up* models to larger numbers of parameters and training configurations, a very important question is how unconstrained MLIPs behave in this limit. Here we investigate this issue, showing that – when trained on some of the current large-scale datasets – unconstrained models can be competitive in accuracy and superior in speed when compared to physically constrained models. We assess these models both in terms of benchmark accuracy and in terms of usability in practical scenarios, focusing on static simulation workflows such as geometry optimization and lattice dynamics. We conclude that accurate unconstrained models can be applied with confidence, especially given that simple inference-time modifications can be used to recover observables that are fully consistent with the relevant physical symmetries.

## 1 INTRODUCTION

The study of interatomic potentials has long underpinned computational chemistry and materials science, providing a framework for understanding how atoms interact and how their interactions govern stability, reactivity, and thermodynamic behavior. Interatomic potentials are essential in methods such as Monte Carlo simulations (Metropolis et al., 1953), molecular dynamics (Alder & Wainwright, 1959), and geometry optimization, where they enable mechanistic exploration of atomic-scale processes across molecular, biological, and condensed-matter systems. Traditional potentials rely on physically motivated analytic forms, which are cheap to evaluate, but whose accuracy is fundamentally limited by the simplicity of their mathematical formulation.

The past decade has witnessed the widespread adoption of machine-learned interatomic potentials (MLIPs) (Behler & Parrinello, 2007). By training on reference data from quantum-mechanical calculations, MLIPs achieve accuracy approaching first-principles methods at highly reduced computational cost. While initial developments focused on system-specific training, the availability of increasingly diverse datasets (Deng et al., 2023) – spanning much of the periodic table and containing millions of labeled configurations – has driven the training of general-purpose (or universal) MLIPs (Chen & Ong, 2022), which can afford good-quality predictions across very diverse systems. This shift has encouraged the adoption of more expressive architectures, notably graph neural networks (Duval et al., 2023), and has even prompted some models to discard explicit physical symmetries in favor of unconstrained architectures where they are learned from the training data (Gasteiger et al., 2021; Pozdnyakov & Ceriotti, 2023). While unconstrained architectures can achieve high computational efficiency, most often by not enforcing rotational symmetries and/or conservation of energy, most models trained for practitioners, as well as for popular benchmarks (Riebesell et al., 2023), rely on explicitly enforcing all physical symmetries of the learning target.

In this work, we show that fully unconstrained architectures can be scaled to large and diverse datasets, achieving accuracies comparable to state-of-the-art equivariant neural networks. We find

that unconstrained models tend to be very efficient at inference time – a very desirable property in molecular simulations – although they often require a large number of epochs to train from scratch, as they need to infer from the data the existence of symmetries and conservation laws. We also demonstrate their applicability to structural optimization and lattice dynamics, showing that the potential pitfalls of unconstrained models can be corrected with minimal effort.

## 2 BACKGROUND AND RELATED WORK

### 2.1 INTERATOMIC POTENTIALS AND THEIR PROPERTIES

An interatomic potential is a function describing the energy of an atomic structure:

$$V(\{\boldsymbol{r}_i, a_i\}_{i=1}^N), \tag{1}$$

where $\boldsymbol{r}_i$ are the three-dimensional positions of the atoms, $a_i$ are their atomic types (chemical elements), and $i$ is an index running over the $N$ atoms in the structure.

**E(3)-invariance.** Interatomic potentials are invariant under the transformations of the Euclidean group in three dimensions E(3), which includes translations, rotations and reflections. Given a group element $g \in E(3)$ acting on all positions, then $V(\{g \cdot \boldsymbol{r}_i, a_i\}_{i=1}^N) = V(\{\boldsymbol{r}_i, a_i\}_{i=1}^N)$.

**Permutational invariance.** Interatomic potentials are invariant with respect to permutations of atom indices, i.e., $V(.., \boldsymbol{r}_i, a_i, .., \boldsymbol{r}_j, a_j, ..) = V(.., \boldsymbol{r}_j, a_j, .., \boldsymbol{r}_i, a_i, ..)$.

**Locality.** With few exceptions, atoms that are distant from one another affect the potential energy function independently (in other words, they do not interact (Prodan & Kohn, 2005)). Mathematically, if atoms $m$ and $n$ are distant (i.e., $|\boldsymbol{r}_m - \boldsymbol{r}_n|$ is large), then $V(.., \boldsymbol{r}_m, a_m, .., \boldsymbol{r}_n, a_n, ..) \approx V'(\{\boldsymbol{r}_i, a_i\}_{i=1, i \neq n}) + V''(\{\boldsymbol{r}_i, a_i\}_{i=1, i \neq m})$, for some functions $V'$ and $V''$.

### 2.2 MACHINE-LEARNED INTERATOMIC POTENTIALS

Machine-learned interatomic potentials (Behler & Parrinello, 2007; Unke et al., 2021) (MLIPs) address the long-standing trade-off between accuracy and efficiency in atomistic simulations. Classical empirical potentials are computationally cheap, but their simple analytic forms severely limit transferability and predictive power, and they are only available for a few selected systems. Conversely, quantum-mechanical methods such as density functional theory (DFT) provide highly accurate energies and forces, but their cost restricts applications to small systems or short timescales. MLIPs provide a middle ground: by training flexible function approximators on quantum-mechanical data, they achieve near–first-principles accuracy while maintaining orders-of-magnitude lower computational cost. Graph neural networks (GNNs) have emerged as particularly effective architectures for MLIPs, as they are well-suited to incorporate essential physical symmetries such as permutational and translational invariance, as well as locality by virtue of constructing graph edges based on a cutoff radius. As a result of these aspects and due to their expressivity, GNNs can achieve high accuracy and generalization across chemical systems, explaining their prevalence in current benchmarks (Riebesell et al., 2023) and practical applications.

While early MLIPs were tailored to specific systems (Deringer et al., 2021; Unke et al., 2021; Behler, 2021), recent years have seen the development of universal MLIPs (Chen & Ong, 2022; Deng et al., 2023; Batatia et al., 2023; Bochkarev et al., 2024; Rhodes et al., 2025; Mazitov et al., 2025a; Park et al., 2024; Wood et al., 2025) trained on increasingly large and diverse datasets Deng et al. (2023); Schmidt et al. (2024); Barroso-Luque et al. (2024); Levine et al. (2025); Chanussot et al. (2021); Tran et al. (2023); Eastman et al. (2023); Mazitov et al. (2025b). Universal MLIPs promise to become general-purpose tools for computational materials science and chemistry: pretrained on massive datasets, they can then be applied directly or after inexpensive fine-tuning for downstream tasks. The success of universal MLIPs demonstrates that generalization across diverse systems is attainable and motivates the exploration of increasingly expressive architectures and scaling strategies. In particular, there is growing interest in using functional forms that do not enforce the physical properties of the potential, but learn them from the data, as will be discussed further in Sec. 3.1.

**Interatomic potentials and forces** Most applications (including molecular dynamics, geometry optimization, phonon calculations; see Sec. 2.3) make use predominantly or exclusively of interatomic forces, as opposed to potential energies. In this context, forces can be defined as the negative derivative of the potential energy function $V$ with respect to the atomic positions:

$$\boldsymbol{f}_j = -\partial V(\{\boldsymbol{r}_i, a_i\}_{i=1}^N) \big/ \partial \boldsymbol{r}_j, \tag{2}$$

where $\boldsymbol{f}_j$ denotes the force acting on atom $j$. Although most MLIPs provide forces by automatic differentiation through Eq. 2, several recent state-of-the-art MLIPs are trained and/or provide inference using "direct" forces (Neumann et al., 2024; Qu & Krishnapriyan, 2024; Rhodes et al., 2025; Fu et al., 2025; Wood et al., 2025), i.e., forces that are simply predicted as an additional head of the model. By avoiding a backward differentiation step, this provides a speedup of a factor between 2 and 3, depending on the architecture (Bigi et al., 2024), as well as reduced memory usage. Since forces that do not obey Eq. 2 are not guaranteed to conserve energy in simulations, they are also referred to as *non-conservative* forces. In contrast, forces obeying Eq. 2 are also called *conservative*.

## 2.3 APPLICATIONS AND BENCHMARKS

MLIPs are routinely used for a wide range of applications. Conceptually, the most simple are geometry optimization and lattice dynamics calculations. Given a set of atomic types and initial positions, geometry optimization finds a nearby local minimum of the potential energy surface as a function of the atomic positions – with low-energy minima being candidates for thermodynamically-stable configurations. A more rigorous assessment of stability can be achieved by a *convex hull* construction, that consists of collecting several local minima, differing by structure or composition, and determining those that have lower (free) energy than a mixture of other phases with the same overall density or chemical composition. To also incorporate finite-temperature effects one can perform *lattice dynamics* (phonon) calculations, which involve computing the mass-scaled Hessian of the potential around a minimum, and diagonalizing it to obtain its eigenvalues (vibrational frequencies) and eigenvectors (the corresponding atomic displacements). From lattice vibrations one can evaluate harmonic free energy corrections to the potential energy of the minima, as well as information on dynamical properties such as infrared or Raman spectra; anharmonic effects can be included considering phonon-phonon interactions via perturbation theory (Peierls, 1929). Alternatively, exact anharmonicity can be obtained by explicitly sampling the Boltzmann distribution $e^{-V/k_B T}$, which can be done through Monte Carlo (Metropolis et al., 1953) or molecular dynamics (Alder & Wainwright, 1959; Stillinger & Rahman, 1974; Andersen, 1980) simulations. Sampling based on the potential energy allows the computation of quantities such as order–disorder transition temperatures, phase diagrams, or specific heats. Between the two, molecular dynamics simulations are often preferred for their superior computational efficiency for medium-sized and large systems (Allen & Tildesley, 2017). Given that most of these applications rely heavily on interatomic *forces*, rather than only the potential, direct force evaluation can speed up almost every MLIP workflow.

Even though the ultimate test of the utility of a MLIP is whether it can be used reliably for practical atomistic simulation tasks, it is often useful to have standardized benchmarks to encourage the development of more accurate and/or efficient MLIPs. Matbench-discovery (Riebesell et al., 2023) was among the first to be introduced, and it features the largest number of state-of-the-art architectures, but it does not contain tests on molecular dynamics applications. Very recently, LAMbench (Peng et al., 2025) and MLIP Arena Chiang et al. (2025) have also been proposed. These benchmark suites focus almost exclusively on materials, and no analogues exist for molecules, to the best of our knowledge. As a result, practitioners often evaluate model accuracy on the various test splits of the SPICE-MACE-OFF dataset (Eastman et al., 2023; Kovács et al., 2025) to benchmark models for applications in organic chemistry and biochemistry.

## 3 THEORY AND METHODS

The question of whether a physical prior should be built into the functional form of a MLIP or learned as part of the training process is not one with a simple answer. The best strategy depends on how constraining the functional form affects the expressivity of the architecture, as well as its computational cost – which in turn depends on how well the necessary mathematical operations are supported by modern libraries and hardware. It also important to consider how hard it is to learn the

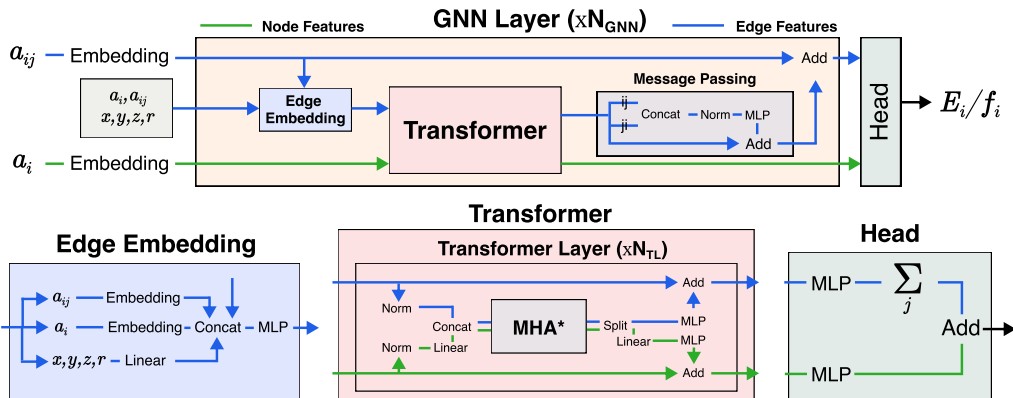

Figure 1: Illustration of the proposed architecture. $a_i$ and $a_{ij}$ are the chemical elements of a center atom and a neighbor atom, respectively. $E_i$ represents an atomic energy; all atomic energies are summed to obtain the total energy.
*Attention weights are scaled to ensure smoothness as described in Pozdnyakov & Ceriotti (2023).

symmetry, and to monitor and correct for it at inference time. Looking at the current landscape, modern GNN-based MLIPs almost universally enforce permutation and translation symmetries, while several recent models relax the requirement for rotational symmetry and energy conservation.

### 3.1 WHAT DETERMINES WHETHER A SYMMETRY IS LEARNABLE?

One simple and convenient way to learn symmetries is to apply data augmentation during training. The difficulty in learning a symmetry can then be estimated in terms of the number of transformed structures that must be generated to provide symmetry information on a resolution comparable to the scale over which one can expect substantial changes in the learned interatomic potential, which we roughly estimate at about 0.1Å for MLIP applications.

**Translations.** Learning the translational invariance of the energy can be considered impossible, as one would need to perform an infinite number of augmentations. Fully periodic structures, however, require only a finite number of translations within our assumptions: for a 10 Å × 10 Å × 10 Å cell, one would need approximately (10 Å / 0.1 Å)$^3 \approx 10^6$ augmentations. Besides the fact that this number of augmentations is too large to be used, the resulting potential energy surface would not be transferable to ranges of Cartesian coordinates outside the training set. In practice, translational invariance is often achieved as a byproduct of enforcing locality, which further explains why it is applied almost universally.

**Permutations.** Considering a small number of neighbors of 30 within a single atomic environment, one would have to augment each structure 30! times to train a permutationally unconstrained model, which is unfeasible. Examples of models using random sampling of permutations appeared in the early days of the field (Montavon et al., 2012), but modern GNNs almost invariably enforce permutation invariance by making use of permutationally invariant pooling operations.

**Rotations.** If we consider the furthest atom that contributes significantly to the description of an atomic environment within a GNN layer to be distant around 4 Å from the central atom, a rotationally unconstrained model would have to learn invariance over a spherical surface of $4\pi(4\text{Å})^2$, yielding $4\pi(4\text{Å})^2/(0.1\text{Å})^2 \approx 20\,000$ augmentations, which is attainable, especially for large datasets which contain similar environments with different orientations, and which therefore contribute to the sampling of rotational symmetry. It is also relatively easy to reduce symmetry breaking by averaging over a grid of rotations at inference time, which makes it practical to relax this symmetry constraint.

**Inversions.** Data augmentation for the inversion symmetry only involves one additional structure, for a total of two. The low information requirement of this symmetry results in equivariant architectures often attempting to alleviate its cost, either by excluding pseudotensor representation from the

neural network (Batzner et al., 2022; Batatia et al., 2022; Musaelian et al., 2023), potentially losing expressivity, or by not enforcing inversion symmetry entirely (Liao et al., 2023; Frank et al., 2022; Fu et al., 2025), resulting in SE(3)-invariant predictions as opposed to E(3)-invariant predictions.

**Energy conservation.** To the best of our knowledge, there is no simple way to formulate a conservative constraint in terms of data augmentation. The energy conservation condition is equivalent to that of a symmetric Jacobian, which is a square matrix of size $3N$, for a total of 8010 constraints for a typical 30-atom training structure. One way to probe this type of symmetry breaking, which could be used as a penalty term during training, consists of checking for the symmetry of the Jacobian of the forces (i.e., the Hessian of the potential for an energy-conserving model). However, this is too expensive to be done in practice, and non-conservative models rely on the presence of geometrically close structures in the dataset to learn approximate energy conservation.

## 3.2 ARCHITECTURE

In order to explore the feasibility of training on large datasets in a fully rotationally unconstrained fashion, we choose the PET architecture (Pozdnyakov & Ceriotti, 2023), a GNN whose successive layers process the edges within each atomic neighborhood through a standard transformer, where each token corresponds to one edge. To make the architecture and training protocol better adapted to larger-scale datasets, we apply some changes with respect to Pozdnyakov & Ceriotti (2023): (1) We use a more modern transformer architecture, including root mean square layer normalization (Zhang & Sennrich, 2019), the SwiGLU (Shazeer, 2020) activation function, and pre-normalization (Xiong et al., 2020). (2) We increase by a factor of four the number of node features for a given edge feature size (the two are instead the same in Pozdnyakov & Ceriotti (2023)). Indeed, processing node features is extremely cheap compared to edge features, as there are typically 30-50 times more edges than nodes in a chemical structure with typical cutoff radii. This allows one to greatly increase the parameter counts of the models while producing nearly zero overhead on training and inference timings and memory usage. This change is applicable to nearly all GNN-based MLIP architectures. (3) We do not discard node features at every GNN layer, but pass them to the next. This is important in architectures with a larger number of GNN layers (which are used in this work) and where the node features constitute a larger fraction of the total representation power of the neural network. (4) We change the learning schedule to a standard linear warm-up followed by cosine decay, as opposed to successive learning rate reductions. (5) Direct forces can be predicted by an additional head that outputs a 3-vector for each atom, similar to how atomic energies are predicted. A more comprehensive description of the architecture is available in Fig. 1 and App. A.

## 3.3 OBTAINING PHYSICAL OBSERVABLES FROM UNCONSTRAINED MODELS

While obtaining physically accurate observables from unconstrained models has been demonstrated in some applications, this practice has not been established in general. Small amounts of symmetry breaking usually do not quantitatively affect the predicted properties (e.g., the relative stability of different phases or compositions when performing a convex-hull construction). However, some care is required to avoid qualitative changes in results, especially within automated workflows. It is worth remembering that the practical implementations of many electronic-structure methods often introduce similar artifacts: for instance, real-space grids break translational symmetry, an issue that is still actively worked on (Durham et al., 2025). For symmetries associated with a compact group, it is always possible to reduce or eliminate the symmetry error by inference-time augmentation, e.g., summing over a Lebedev grid (Langer et al., 2024), or by ensembling (Gerken & Kessel, 2024).

**Molecular dynamics.** Non-equivariant models can often be used out of the box in molecular dynamics simulations with minimal downsides (Langer et al., 2024; Mazitov et al., 2025a). Systematic long-time errors can be avoided by evaluating the MLIP and its forces on a different random rotation (and inversion) of the structure at every time step (Langer et al., 2024), with negligible overhead. Direct force models can also recover quantitative physical observables in molecular dynamics (Bigi et al., 2024) using a multiple time-stepping algorithm (Tuckerman et al., 1992) where conservative and non-conservative forces are used together, with the computationally advantageous non-conservative forces constituting the majority of the evaluations.

**Geometry optimization.** Geometry optimization with non-conservative force models was investigated in Bigi et al. (2024), showing that, although inaccurate direct-force models struggle to converge, accurate models have nearly no downsides. This is intuitive, as the target force field is conservative, and therefore the model must be conservative at least up to its force error. A similar investigation on the effects of using rotationally unconstrained models for geometry optimization (Sec. 4.1) shows that symmetry breaking is possible but benign, and possibly even beneficial as it can relax away from unstable high-symmetry structures. It might however be necessary to use more permissive thresholds to identify the symmetry of the relaxed structure than when using exactly equivariant models. Whenever the space group is known, it is also possible to restore exact symmetry by projecting out the component of the forces incompatible with the group action, or by averaging the model predictions over the symmetries of the group.

**Frequency and phonon calculations.** Phonon calculations often assume exact symmetry of the underlying structure, since phonon bands are plotted along high-symmetry lines in the lattice Brillouin zone (BZ). This makes it especially important to either perform a constrained-symmetry optimization, relax the symmetry-detection threshold, or fix the BZ sampling path to that of the high-symmetry group. The residual stress left after symmetry-constrained relaxation due to the lack of equivariance is sufficiently small that phonons can be computed at the symmetric minimum without artifacts such as spurious soft modes. At the unconstrained, slightly distorted minimum, BZ-integrated quantities like the phonon density of states are practically indistinguishable from the symmetric case, indicating that the absence of exact equivariance has little impact on global properties. Direct-force models require additional care, as they do not guarantee a vanishing total force, and their Jacobian with respect to atomic displacements is not the (symmetric) Hessian of a potential energy function. In practice, both conditions are easy to meet: the net force is manually subtracted from the predictions, and the Jacobian is symmetrized by default in most phonon codes.

## 4 RESULTS

### 4.1 LARGE-SCALE MATERIALS DATASETS

We apply the proposed architecture to three large-scale materials databases – the MPtrj dataset (Deng et al., 2023), the subsampled Alexandria dataset (Schmidt et al., 2024; Barroso-Luque et al., 2024) and OMat24 (Barroso-Luque et al., 2024) – to demonstrate its behavior in the large-data regime and to compare it to state-of-the-art equivariant MLIPs. The PET-MPtrj model contains 260M parameters, while PET-OAM (trained on all three datasets) contains 190M parameters. Thanks to the increased size of the node representations and the unconstrained architecture, these relatively large parameter counts can be achieved without sacrificing computational efficiency (see inference timings in Sec. 4.2). While larger models would have almost certainly performed better, especially in the case of PET-OAM, we decided to limit the size of the neural networks to make training (and inference) more affordable. For the same reason, all presented models use a cutoff radius of 4.5 Å, as opposed to the standard choice of 6.0 Å for these datasets. More details are available in App. A.

Both models were pre-trained using non-conservative forces and stresses, then fine-tuned using their conservative counterparts. This two-step procedure was shown to save computational time in Bigi et al. (2024) and Fu et al. (2025), occasionally also resulting in better accuracies. Compared to other published training protocols (Park et al., 2024; Bochkarev et al., 2024; Fu et al., 2025), we find that our rotationally unconstrained architecture takes more passes through the data to train to convergence (although this does not necessarily translate to a higher computational cost, see Sec. 4.2). This is not surprising: as argued in Sec. 3, an unconstrained model needs to learn the equivariance constraints during training. In contrast, trained unconstrained models can be fine-tuned quickly: for example, as shown in App. B, a model trained on OMat24 can be fine-tuned to the smaller MAD dataset in less than 1/20th of the epochs needed to train from scratch, achieving 50% lower test errors.

**Geometry optimization and phonon calculations from unconstrained models.** As a demonstrative example of the potential pitfalls of unconstrained models in static lattice calculations, we consider the case of elemental titanium. The stable phase of Ti at high temperature is body-centered cubic (BCC), but it is stabilized by entropic effects and static calculations find the low-temperature

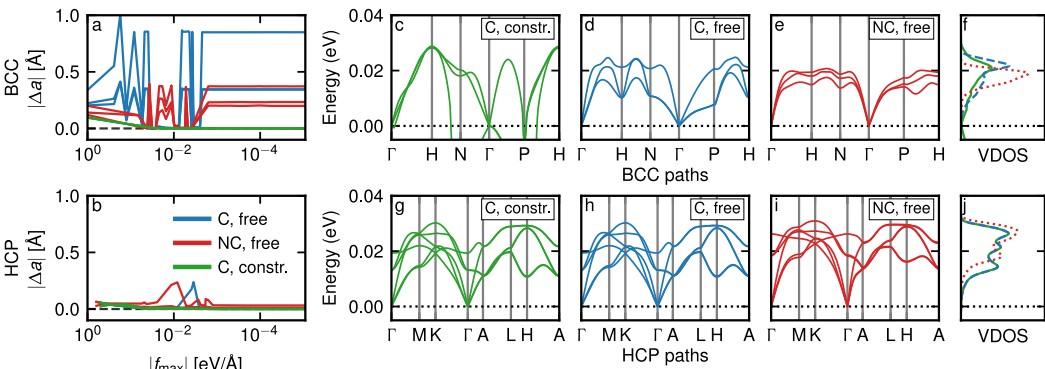

Figure 2: Geometry optimization and phonon calculations with PET-OAM. (a,b) Deviation of unit-cell lengths as a function of the maximum generalized force (force and stress component, as returned by ASE's "FrechetCellFilter") for BCC (a) and HCP (b) starting configurations. (c–e) Phonon bands of titanium starting from BCC, using conservative PET after constrained (c) and unconstrained (d) relaxation, and non-conservative PET after unconstrained relaxation (e). (g–i) Corresponding calculations starting from HCP. (f,j) Vibrational density of states for the two sets of calculations.

hexagonal-closed-packed (HCP) to be preferred. In an equivariant model, symmetric but unstable structures such as BCC exhibit zero forces and scalar stress, so any relaxation initialized in BCC remains trapped within that symmetry. Non-equivariant models are not bound by this constraint, and in fact residual asymmetries drive the relaxation toward more stable close-packed structures—in this case, a face-centered cubic (FCC) structure that can be obtained with a continuous deformation path. Constraining the symmetry by projecting out the incompatible force components conserves the symmetry, for both conservative and non-conservative forces. Due to the presence of a small anisotropy in the stress, free relaxation leads to symmetry breaking—a small one for the stable HCP structure, and a dramatic deformation for the BCC cell, that relaxes into the locally stable FCC geometry (Figure 2). Note that O(3)-averaging reduces but does not eliminate the anisotropy, and so it delays but does not prevent relaxation of the unstable BCC initial configuration. We perform conservative and non-conservative relaxation for the 256 963 structures within the WBM dataset (Wang et al., 2021), with a convergence threshold of 0.05 eV/Å. We find that 64% (59%) of the final configurations are detected to retain the symmetry of the the initial structure with a threshold of 0.01 Å for the conservative (non-conservative) model, which is the default, and 89% (89%) with a threshold of 0.1 Å. O(3)-averaging increases the number of symmetric structures to 81% (80%) and 91 % (90 %), respectively. We recommend using rotational averaging and/or a slightly increased tolerance whenever performing geometry optimizations with unconstrained models.

The outcomes of a phonon calculation depend mostly on the relaxed geometry. Using a constrained BCC structure leads (as it should) to unstable phonon modes, whereas the relaxed FCC structure is locally stable and displays no unstable lattice vibrations. When considering the stable HCP configuration, instead, the phonon dispersion curves are almost identical regardless of whether the structure has been free to relax to a slightly symmetry-broken geometry. We reiterate that automatic workflows might show a dramatically different path if a lower symmetry is detected. This is inconvenient, but has no consequence on physical properties such as the total vibrational density of states, that depend on an integral over the entire BZ. Even when symmetrizing the Hessian and removing the total force component, non-conservative models tend to yield significantly different phonon dispersion curves. Increasing the finite-difference displacements used to estimate the Hessian helps stabilize calculations, but at present we consider it to be safer to use a conservative model when evaluating phonons. These demonstrative examples are representative of the general behavior of unconstrained models for lattice dynamics calculations. A conservative, rotationally unconstrained model can be used safely—as will be clear from the excellent performance on benchmarks that use vibrational properties—although care must be taken to avoid qualitative changes in the detected symmetry.

**The matbench-discovery benchmark** Although public benchmarks should not be taken as the sole metric to assess the usefulness of a model, they provide an objective scale on which one

Table 1: Model performance on selected metrics from the matbench-discovery benchmark suite. Besides models presented in this work, the top 6 models from the benchmark leaderboard are also shown. For each metric, the best model is highlighted in bold and the second best is underlined. *Unlike other models, eSEN-MPtrj was trained using denoising non-equilibrium structures (DeNS), which improves accuracy on all tested metrics in Liao et al. (2024), and ORB-OAM uses a diffusion pretraining scheme Rhodes et al. (2025).

| Model | MPtrj | | | | | OAM | | | | |
|---|---|---|---|---|---|---|---|---|---|---|
| | DAF($\uparrow$) | Acc.($\uparrow$) | F1($\uparrow$) | $\kappa_{SRME}$($\downarrow$) | RMSD($\downarrow$) | DAF($\uparrow$) | Acc.($\uparrow$) | F1($\uparrow$) | $\kappa_{SRME}$($\downarrow$) | RMSD($\downarrow$) |
| PET | 5.251 | 0.939 | 0.804 | 0.527 | 0.079 | **6.089** | 0.971 | 0.902 | 0.203 | 0.065 |
| eSEN* | **5.260** | **0.946** | **0.831** | **0.340** | **0.075** | 6.069 | **0.977** | **0.925** | 0.170 | **0.061** |
| NequIP | 4.704 | 0.921 | 0.761 | 0.452 | 0.086 | 5.823 | 0.967 | 0.893 | **0.166** | 0.065 |
| ORB* | 4.702 | 0.922 | 0.765 | 1.725 | 0.101 | 5.912 | 0.971 | 0.905 | 0.210 | 0.075 |
| SevenNet | 4.629 | 0.920 | 0.760 | 0.550 | 0.085 | 5.825 | 0.969 | 0.901 | 0.317 | 0.064 |
| Allegro | 4.516 | 0.915 | 0.751 | 0.504 | 0.082 | 5.674 | 0.966 | 0.895 | 0.319 | 0.065 |
| GRACE | 4.163 | 0.896 | 0.691 | 0.525 | 0.090 | 5.774 | 0.963 | 0.880 | 0.294 | 0.067 |

can compare different architectures. Among material-based benchmarks for MLIPs, matbench-discovery (Riebesell et al., 2023) features the widest variety of architectures. This benchmark emphasizes metrics that reflect the ability of a model to be used for materials discovery, and especially its ability to detect the most stable configurations among those of similar density or composition through a convex hull analysis. We focus on the following metrics, which are useful predictors of performance in material discovery workflows: (1) DAF (discovery acceleration factor): the acceleration factor of material discovery through the model compared to brute-force search; (2) accuracy: this is the fractional accuracy of the exercise if seen as a binary classification task (stable/unstable); (3) F1 score: harmonic mean of precision and recall for the binary classification task. Besides these three metrics, we will also use (4) an RMSD metric for geometry optimization (root-mean-square deviation of the MLIP-optimized structure compared to DFT) and (5) $\kappa_{SRME}$, which quantifies the quality of phonons calculated from the model (Póta et al., 2024). The results (Table 1) show that the methods proposed in the current work reach state-of-the-art accuracy on materials discovery workflows, while being competitive with established GNNs for the calculation of static properties at local minima (phonons and optimized geometries). More details are available in App. C. Further benchmarking on LAMbench (Peng et al., 2025) and MADBench (Mazitov et al., 2025a) is available in App. D and E, confirming the very strong performance of the presented models on materials.

## 4.2 THE SPICE MOLECULAR BENCHMARK

Unlike the material domain, applications of MLIPs to molecular systems have not yet seen the development of established benchmark suites. One of the most widely used large-scale MLIP benchmark in the molecular domain is the test accuracy on the SPICE (Eastman et al., 2023) dataset, in the version proposed by Kovács et al. (2025). We train a 190M-parameter model on SPICE, using con-

Table 2: Test set accuracies for models trained on the dataset presented in Kovács et al. (2025). Energy MAEs are shown in units of meV per atom; force MAEs are shown in units of meV/Å. For each metric, the best model is highlighted in bold and the second best is underlined.

| Subset | MACE | | EScAIP | | eSEN | | PET | |
|---|---|---|---|---|---|---|---|---|
| | E | F | E | F | E | F | E | F |
| PubChem | 0.88 | 14.75 | 0.53 | 5.86 | 0.15 | 4.21 | **0.09** | **3.53** |
| DES370K Monomers | 0.59 | 6.58 | 0.41 | 3.48 | 0.13 | 1.24 | **0.10** | **1.00** |
| DES370K Dimers | 0.54 | 6.62 | 0.38 | 2.18 | 0.15 | 2.12 | **0.12** | **1.20** |
| Dipeptides | 0.42 | 10.19 | 0.31 | 5.21 | 0.07 | 2.00 | **0.05** | **1.55** |
| Solvated Amino Acids | 0.98 | 19.43 | 0.61 | 11.52 | 0.25 | **3.68** | **0.17** | 4.37 |
| Water | 0.83 | 13.57 | 0.72 | 10.31 | 0.15 | **2.50** | **0.13** | 3.05 |
| QMugs | 0.45 | 16.93 | 0.41 | 8.74 | 0.12 | **3.78** | **0.08** | **2.91** |

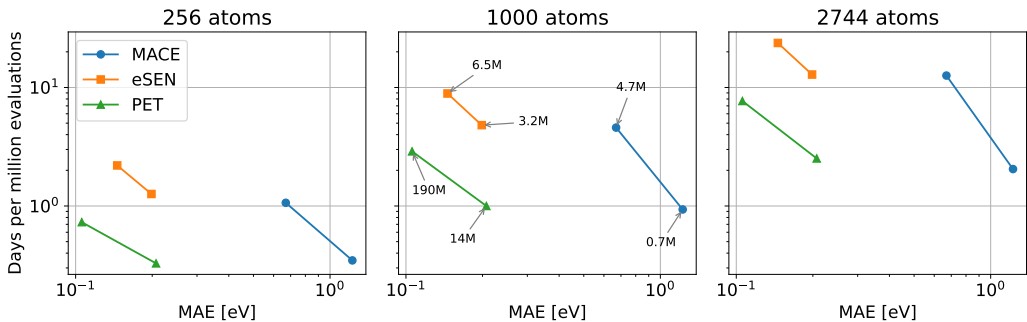

Figure 3: Accuracy-speed Pareto front for models trained on the SPICE dataset, at varying structure sizes. The energy error on the x-axis is the average of all subset-specific errors. Model sizes for the six benchmarked models are shown in the middle panel. More details are available in App. F.

servative forces throughout and training for three times the number of epochs reported by eSEN (Fu et al., 2025). Although training times for the latter are not available, the inference timings in Fig. 3 would suggest that the two models required around the same amount of compute to train. The proposed unconstrained architecture exceeds the accuracy of state-of-the-art models on this benchmark (Table 2), demonstrating its remarkable accuracy in the molecular domain.

**Inference timings** Molecular force fields are often used to perform molecular dynamics simulations, making inference efficiency of primary importance. Unlike high-throughput workflows, molecular dynamics is not trivially parallelizable. Furthermore, large-scale simulations are often needed in this domain to fully capture the structural complexity of biochemical processes, increasing the computational cost of inference. Fig. 3 shows three accuracy-speed Pareto fronts for the models presented in Table 2 for systems of varying numbers of atoms. From these experiments, it can be seen that the unconstrained models presented in this work constitute an excellent compromise between accuracy and inference speed. More details on this benchmark are available in App. F.

## 5 DISCUSSION

In this work, we have shown that unconstrained models (in the rotational sense and/or in the direct-force sense) can be scaled successfully to train on large datasets and afford accuracies comparable to state-of-the-art equivariant architectures for MLIPs. From our investigation it appears that, when compared to physically constrained models, non-equivariant models must train for a larger number of epochs to converge training, although the lower cost largely compensates for that. Non-conservative models, on the other hand, show accelerated training (as also observed in Bigi et al. (2024) and Fu et al. (2025)) and can be fine-tuned conservatively for further computational savings.

At inference time, we have demonstrated that rotationally unconstrained models can be more efficient than equivariant models, although further tuning of either class of architectures, as well as software optimization, might shift the balance in both directions. Non-conservative models consistently show the expected 2-3x theoretical speed-up over their conservative counterparts. Even though we were able to establish that rotationally unconstrained models can match the predictive accuracy of equivariant models on downstream tasks (at times with minor modifications to the evaluation procedure), non-conservative force models with equal or better test set accuracies compared to conservative models often fail to achieve the same quantitative performance. We conclude that they are best used as a pre-training step in the training of conservative models (Bigi et al., 2024; Fu et al., 2025; Wood et al., 2025) or in combination with conservative models for finite-temperature applications (Bigi et al., 2024). In addition to making new state-of-the-art models available for practitioners in material discovery and molecular simulations, our work establishes rotationally unconstrained architectures as promising alternatives to achieve better trade-offs between accuracy, parameter budget, and computational cost in general-purpose MLIPs and their applications. In this domain, this class of models has received much less attention, and we believe there is substantial potential for further improvements in both accuracy and efficiency.

ETHICS STATEMENT

This work adheres to the ethical standards expected at ICLR 2026. All experiments were conducted using publicly available datasets under their respective licenses. It is difficult to estimate the societal impact of this work. To the best of our knowledge, atomic-scale modeling techniques are overwhelmingly used for constructive purposes (drug discovery, modeling of new materials) and we would expect the outcomes to have an overall positive impact. We encourage practitioners applying our methods to consider fairness and transparency in their use.

REPRODUCIBILITY STATEMENT

We include information to reproduce the experiments performed in this work in the Appendices. The necessary code and comprehensive instructions are available as Supplementary Material.

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

## A  MODEL AND TRAINING DETAILS

### A.1  ARCHITECTURE

The proposed architecture is based on a modified version of the implementation of PET (Pozdnyakov & Ceriotti, 2023) in metatrain (Bigi et al., 2025).

Besides the changes to the architecture described in the main text, a number of further smaller modifications were made, namely:

- the use of skip connections instead of summing layer-wise predictions
- the use of more processed representations to make predictions
- a multi-layer perceptron is used to combine $ij$ and $ji$ representations during message passing, as opposed to a plain sum

These, along with the changes highlighted in the main text, are illustrated in Fig. 1. All proposed modifications improve the accuracy of training on diverse datasets. As an example, we report a 10-15% improvement in force MAE on the MAD dataset in Fig. 4 compared to literature results, with comparable computational efficiency. However, we do not exclude that further tuning of the architecture might improve accuracies and/or inference timings further.

### A.2  TRAINING

All training and fine-tuning runs were executed with a standard cosine learning rate scheduler, with a linear warm-up stage corresponding to 10% of the total number of training steps. All training runs were executed in single-precision, with the non-conservative pre-training runs (OMat-NC and MPtrj-NC in Tab. 3) using TensorFloat-32 operations. A chemical-composition-based linear model was fitted for the energies and removed from the energy targets before all fitting exercises. Furthermore, scaling of all targets to unit standard deviation across the training set was employed. A standard Huber loss was used for all training runs, except for fine-tuning on the MAD dataset (Sec. B), where a MSE loss was used. The total loss is composed of a sum of individual terms for per-atom energies, forces and per-atom virials (we use stresses instead in the non-conservative case). Rotational and inversion-based data augmentation was employed for all training runs, according to the symmetry of the different targets (energy, forces, stress/virial). Gradient norm clipping to a value of 1.0 was used throughout. It should be noted that all conservative fine-tuning runs keep training the non-conservative force and stress outputs with a small weight (0.01) in the loss function. This is done to allow multiple-time-stepping simulations using both conservative and non-conservative heads (Bigi et al., 2024).

The official training/validation/test splits were used for the OMat24 (Barroso-Luque et al., 2024), sAlex (Barroso-Luque et al., 2024), SPICE (Kovács et al., 2025) and MAD (Mazitov et al., 2025b) datasets. A random 80:10:10 split was used for the MPtrj dataset (Deng et al., 2023), which does not have official splits as far as we are aware.

### A.3  HYPERPARAMETERS

The hyperparameters employed to train the models presented in this work are shown in Tab. 3.

Table 3: Model and training hyperparameters for the models evaluated in this work. A linear learning-rate warm-up of 10% of the total number of epochs was performed. The number of node features corresponds to four times that of edge features for all models.

| Model | OMat-NC | OMat-C | OAM-NC | OAM-C | MPtrj-NC | MPtrj-C | SPICE-L | SPICE-S | OMAD |
|---|---|---|---|---|---|---|---|---|---|
| Training starts from | scratch | OMat-NC | OMat-NC | OMat-C | scratch | MPtrj-NC | scratch | scratch | OMat-C |
| Parameter count | 190M | 190M | 190M | 190M | 280M | 280M | 190M | 14M | 190M |
| Conservative training | no | yes | no | yes | no | yes | yes | yes | yes |
| Number of epochs | 25 | 5 | 25 | 1 | 300 | 200 | 300 | 300 | 30 |
| Edge features | 512 | 512 | 512 | 512 | 384 | 384 | 512 | 192 | 512 |
| GNN layers | 3 | 3 | 3 | 3 | 3 | 3 | 3 | 3 | 3 |
| Attention layers | 2 | 2 | 2 | 2 | 5 | 5 | 2 | 1 | 2 |
| Graph cutoff radius (Å) | 4.5 | 4.5 | 4.5 | 4.5 | 4.5 | 4.5 | 4.5 | 4.5 | 4.5 |
| Max learning rate | 2e-4 | 1e-4 | 5e-5 | 5e-5 | 2e-4 | 1e-4 | 2e-4 | 2e-4 | 5e-5 |
| Batch size | 2048 | 1024 | 256 | 256 | 256 | 128 | 128 | 128 | 8 |
| E loss weight | 1.0 | 1.0 | 1.0 | 1.0 | 1.0 | 1.0 | 1.0 | 1.0 | 1.0 |
| F loss weight | 1.0 | 1.0 | 1.0 | 1.0 | 1.0 | 1.0 | 1.0 | 1.0 | 1.0 |
| S/V loss weight | 0.1 | 1.0 | 1.0 | 1.0 | 0.1 | 1.0 | – | – | 1.0 |
| E Huber threshold | 0.015 | 0.015 | 0.010 | 0.010 | 0.010 | 0.010 | 0.003 | 0.003 | – |
| F Huber threshold | 0.010 | 0.040 | 0.050 | 0.050 | 0.025 | 0.025 | 0.004 | 0.004 | – |
| S/V Huber threshold | 0.004 | 0.030 | 0.005 | 0.050 | 0.004 | 0.050 | – | – | – |

## B  FINE-TUNING ON SMALLER DATASETS

As a demonstration of the fine-tuning capabilities of unconstrained architectures, we consider fine-tuning of models trained on the OMat24 (Barroso-Luque et al., 2024) dataset. We fine-tune this model on the MAD (Mazitov et al., 2025b) dataset, which is smaller by more than 1000 times, but which was nonetheless used to train a successful general-purpose interatomic potential in Mazitov et al. (2025a). We compare the fine-tuned model with training from randomly initialized weights on the same dataset and with the reported accuracy from Mazitov et al. (2025a) in Fig. 4.

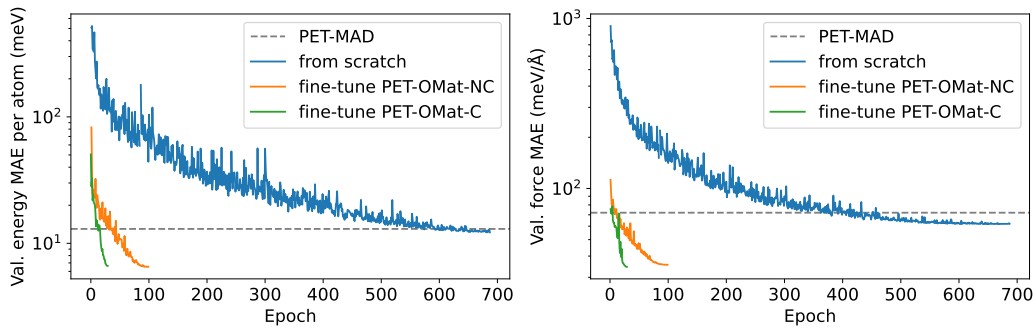

Figure 4: Validation set accuracy of the PET-MAD (Mazitov et al., 2025a) model, a model trained from scratch on the MAD dataset using the architecture proposed in this work, and fine-tuning from a model trained on the OMat24 dataset.

Fine-tuning our OMat models leads to halving of the validation energy and force errors compared to training from scratch. Furthermore, training time is greatly reduced: we found that it is possible to achieve near-converged validation metrics by fine-tuning for 100 epochs on a non-conservative OMat model and 30 epochs on a conservative OMat model, as opposed to 700 or more when training from scratch. We postulate that, once equivariance is learned on a larger datasets, fine-tuning is efficient and largely unaffected by any potential training-time slowdowns due to learning equivariance. This demonstrates the benefits of using our pre-trained models to achieve better accuracies on smaller datasets, whose targets can be computed consistently at high levels of theory (i.e., using more expensive approximations of quantum mechanics).

In Sec. E, we show that this "OMAD" model, when evaluated consistently, shows superior accuracy and transferability compared to OAM models.

## C MATBENCH-DISCOVERY EVALUATION

Evaluation on matbench-discovery (Riebesell et al., 2023) is executed using standard scripts, which are available in the Supplementary Material and are similar to those of other published models. We employ O(3)-averaging over a Lebedev grid with L=3 for the geometry optimization and material discovery tasks, while we do not find symmetrization to have an impact on the phonon-related task in the benchmark suite and we use our unsymmetrized models as a result.

**Non-conservative models** Tab. 4 contains a comparison between the metrics achieved by conservative and non-conservative models on this benchmark. Although we were able to obtain relateively good metrics from non-conservative models (especially on $\kappa_{\text{SRME}}$ when compared to the literature, see also the next paragraph), we find that non-conservative models consistently perform worse on the downstream tasks considered in matbench-discovery, despite them often achieving better training and validation accuracies.

Table 4: Performance comparison of conservative and non-conservative models on the matbench-discovery benchmark.

| Model | MPtrj | | | | | OAM | | | | |
|-------|-------|------|------|-------|-------|------|------|------|-------|-------|
| | DAF($\uparrow$) | Acc.($\uparrow$) | F1($\uparrow$) | $\kappa_{\text{SRME}}(\downarrow)$ | RMSD($\downarrow$) | DAF($\uparrow$) | Acc.($\uparrow$) | F1($\uparrow$) | $\kappa_{\text{SRME}}(\downarrow)$ | RMSD($\downarrow$) |
| PET-C | 5.251 | 0.939 | 0.804 | 0.527 | 0.079 | 6.089 | 0.971 | 0.902 | 0.203 | 0.065 |
| PET-NC | 4.706 | 0.920 | 0.755 | 0.995 | 0.089 | 5.989 | 0.967 | 0.889 | 0.623 | 0.083 |

**Effect of the finite-difference displacement on phonon calculations** Tab. 5 reports the performance of different models on the phonons task of matbench-discovery, which evaluates the symmetric mean relative error on lattice thermal conductivity (Póta et al., 2025), for varying finite-difference displacements in force-constant calculations. All results use central finite differences. We find that this choice is crucial for non-equivariant models (and it is also adopted for ORB models), as it can more than halve the error by averaging out small inconsistencies between symmetry-equivalent displacements. Conservative models achieve their best accuracy with a relatively large displacements (0.1 Å for both OAM and MP models). We believe this to be a consequence of roughness in the potential energy surface, although more work would be necessary to establish the exact cause. A similar investigation in Fu et al. (2025) also found that the 0.03 Å standard is not optimal in general. Overall, conservative models can reach performance close to the state of the art, while our non-conservative models also deliver competitive results, outperforming other direct-force models reported on the leaderboard.

Table 5: Effect of finite-different displacement size on the $\kappa_{\text{SRME}}$ metrics of matbench-discovery. "From C" refers to using the non-conservative head of a conservatively fine-tuned model.

| Model | Finite-difference displacement (Å) | | | |
|-------|------|------|------|------|
| | 0.03 | 0.05 | 0.07 | 0.10 |
| PET-OAM-C | 0.238 | 0.229 | 0.217 | **0.203** |
| PET-OAM-NC | 0.873 | 0.777 | 0.704 | **0.623** |
| PET-OAM-NC (from C) | 0.463 | 0.448 | **0.446** | 0.456 |
| PET-MP-C | 0.734 | 0.678 | 0.616 | **0.527** |
| PET-MP-NC | 1.484 | 1.235 | 1.104 | **0.995** |
| PET-MP-NC (from C) | 1.168 | **1.120** | 1.165 | 1.170 |

## D LAMBENCH EVALUATION

Although matbench-discovery (Riebesell et al., 2023) remains the most popular public benchmark for atomistic models in the materials domain, it is important to consider other available benchmarks to obtain a broader perspective of the performance of various models on a more diverse set of tasks.

We evaluated the performance of our models using the recent LAMBench benchmark (Peng et al., 2025), which focuses on probing the generalizability, inference speed, and stability of universal models in atomistic simulations. When interpreting the results below, **it is important to keep in mind that there is usually no consistency between the DFT settings used to prepare the benchmark data and model training sets**. This lack of consistency can severely affect the quality of predictions because it introduces unavoidable and unlearnable noise in the target data, which cannot be separated from the real approximation errors of the models (more detail on this effect can be found in Mazitov et al. (2025a)). Therefore, the results in this section should only be used for qualitative model assessment. Generally, we recommend benchmarks which use consistent levels of DFT in order to perform quantitative assessments (see Sec. E below).

We first run the force-field generalizability test resolved by three separate categories: Molecules, Inorganic Materials and Catalysis. In each category, the accuracy of each model in predicting energies, forces and stresses (if available) is evaluated across several out-of-domain category-specific datasets, and then the overall performance score is computed based on the averaged normalized error values. All technical details on the score calculation procedure can be found in Peng et al. (2025).

The results of the evaluation are shown in Table 6. Among all models trained in this work, PET-OAM-C demonstrates the best score across all three domains, showing the best overall results on Molecules, and reaching the same accuracy as the DPA-3.1-3M on Inorganic Materials. In the Catalysis domain, both conservative and non-conservative PET-OAM models show second-best result, outperforming all other OAM-trained models in the list. The superior performance of the DPA-3.1-3M model in this case most likely originates from the presence of explicit catalytic data in the training set (Zhang et al., 2024). PET-OMAD performs notably worse compared to PET-OAM, and even falls behind the PET-MPtrj on Inorganic Materials, despite being pre-trained on the large OMat24 dataset. This effect stems from differences in ab initio theory levels used in the MAD dataset (on which PET-OMAD is ultimately fine-tuned) and other datasets, which usually have settings consistent (or nearly consistent) to those in the MPtrj dataset (Deng et al., 2023) or the Alexandria dataset (Schmidt et al., 2024), and therefore suffer much less from the difference in the theory baseline. As we demonstrate in Sec. E, PET-OMAD outperforms other models upon consistent evaluation. A more detailed discussion of the effect of the consistency of DFT settings in model evaluation can be found in Mazitov et al. (2025a).

In addition to force-field generalizability, we also perform the property prediction test, which includes reaction and activation energy prediction in catalytic reactions (OC20-NEB task), elastic constant prediction (elastic task), and conformer energy prediction for a selected set of organic molecules (Wiggle150 task). The results are shown in Tables 7, 8, and 9, respectively. The lack of explicit molecular and surface data in the OAM dataset lead to a dramatic decrease in PET-MP and PET-OAM model accuracy compared to bulk systems. At the same time, almost all PET models show high (and sometimes the best) success rates, which means that, in most cases, their MAE in predicting $E_a$ is less than 0.1 eV. This combination is most likely caused by outliers in the data, which can corrupt the overall MAE values, while preserving the fraction of successful runs on a decent level.

At the same time, PET-OMAD achieves the third-best results in predicting reaction energies and barriers, closely competing with DPA-2.4-7M, and even demonstrating the highest success rate in predicting desorption. The presence of explicit molecular and surface data in the MAD dataset significantly improves the model's ability to describe catalytic reactions. We note that these results are still obtained using inconsistent DFT targets, so there is definitely room for improvement upon consistent evaluation.

The conservative PET-OAM model achieves the best overall accuracy in predicting the shear modulus (Table 8), and competes closely with other universal models in predicting the bulk modulus. Upon comparing the results of the conservative (PET-OAM-C and PET-MPtrj-C) models against their non-conservative analogs (PET-OAM-NC) and (PET-MPtrj-NC), we clearly see that elastic moduli are very sensitive to the lack of energy conservation. This comes as no surprise, as geometry optimization is part of the protocol for calculating elastic moduli, and this it can fail upon having no relaxation constraints and no rotational averaging, as we show in Sections 4.1 and G. MAD-trained models (PET-MAD and PET-OMAD) show comparatively worse results due to a difference in the underlying DFT. This observation also indicates that some of the materials properties (like the elastic moduli) can be actually very sensitive to a choice of DFT flavor and settings.

Table 6: Evaluation of various universal MLIPs on the force-field generalizability test of the LAM-Bench benchmark from Peng et al. (2025). Performances of PET-OAM, PET-MPtrj and PET-OMAD are computed in this work, and other results are reproduced from the reference paper (Peng et al., 2025). The domain-specific score ranging from 0 to 1 is computed based on model errors in predicting energies and forces. For each metric, the best model is highlighted in bold and the second best is underlined.

| Model | Task | | |
| --- | --- | --- | --- |
| | Molecules | Inorganic Materials | Catalysis |
| PET-OAM-C | **0.85** | **0.84** | 0.74 |
| PET-OAM-NC | 0.78 | 0.79 | 0.74 |
| PET-MPtrj-C | 0.80 | 0.83 | 0.48 |
| PET-MPtrj-NC | 0.69 | 0.77 | 0.46 |
| PET-MAD | 0.79 | 0.76 | 0.44 |
| PET-OMAD | 0.81 | 0.78 | 0.58 |
| DPA-2.4-7M | 0.82 | 0.80 | 0.66 |
| DPA-3.1-3M | 0.84 | **0.84** | **0.79** |
| MACE-MP-0 | 0.64 | 0.72 | 0.59 |
| MACE-MPA-0 | 0.66 | 0.79 | 0.62 |
| Orb-v2 | 0.78 | 0.75 | 0.70 |
| Orb-v3 | 0.82 | 0.82 | 0.71 |
| SevenNet-l3i5 | 0.75 | 0.76 | 0.51 |
| SevenNet-MF-ompa | 0.75 | 0.83 | 0.66 |
| MatterSim-v1-5M | 0.74 | 0.82 | 0.59 |
| GRACE-2L-OAM | 0.75 | 0.82 | 0.67 |

Table 7: Performance of different universal models on the OC20-NEB task of the LAMBench benchmark. The mean absolute error (MAE) for reaction energy differences ($\Delta E$) and activation energies ($E_a$) are given in eV. The success rates (in %) for transfer, desorption, and dissociation reactions are calculated as a fraction of the corresponding runs for which the MAE in $E_a$ predictions is less than 0.1 eV. The best results per column are shown in bold, and the second-best are underlined.

| Model | MAE $\Delta E$ ($\downarrow$) | MAE $E_a$ ($\downarrow$) | % Transfer ($\uparrow$) | % Desorption ($\uparrow$) | % Dissociation ($\uparrow$) |
| --- | --- | --- | --- | --- | --- |
| PET-OMAD | 0.331 | 1.289 | 69.14 | **98.43** | 66.46 |
| PET-MPtrj-C | 1.257 | 2.131 | 73.14 | 92.91 | 79.75 |
| PET-MPtrj-NC | 2.030 | 2.874 | **80.00** | 94.49 | **85.44** |
| PET-OAM-NC | 3.439 | 3.879 | 70.29 | 81.89 | 70.89 |
| PET-OAM-C | 4.210 | 4.438 | 62.29 | 85.83 | 72.78 |
| DPA-3.1-3M | **0.234** | **1.203** | 66.86 | 81.10 | 63.92 |
| DPA-2.4-7M | 0.306 | 1.271 | 68.57 | 69.29 | 77.85 |
| PET-MAD | 0.471 | 1.408 | 63.43 | **98.43** | 77.85 |
| SevenNet-l3i5 | 0.546 | 1.423 | 77.71 | 87.40 | 75.95 |
| MACE-MPA-0 | 0.566 | 1.459 | 72.57 | 94.49 | 84.81 |
| MACE-MP-0 | 0.580 | 1.468 | 76.57 | 90.55 | 84.81 |
| GRACE-2L-OAM | 0.704 | 1.583 | 65.14 | 90.55 | 72.15 |
| SevenNet-MF-ompa | 1.278 | 2.070 | 66.86 | 92.91 | 68.35 |
| Orb-v3 | 1.470 | 2.298 | 61.71 | 87.40 | 72.15 |
| Orb-v2 | 1.729 | 2.682 | 66.29 | 78.74 | 74.05 |
| MatterSim-v1-5M | 2.018 | 2.686 | 76.00 | 83.46 | 84.81 |

Finally, our results on the Wiggle150 test show that PET-OAM-NC achieves the best results in predicting the energies of molecular conformers among the trained PET models, and second-best result overall, only falling behind the DPA-3.1-3M model. This can be explained by the presence of explicit molecular data in the DPA-3.1-3M training set. The OAM dataset is designed to primarily describe inorganic materials with no information on molecules. However, the results across the

Table 8: Performance of different universal models in predicting the elastic constants from the LAMBench benchmark. The mean absolute errors (MAE) for shear modulus ($G_{\mathrm{VRH}}$) and bulk modulus ($K_{\mathrm{VRH}}$) are given in GPa. The best results per column are shown in bold, and the second-best are underlined.

| Model | MAE $G_{\mathrm{VRH}}$ | MAE $K_{\mathrm{VRH}}$ |
|---|---|---|
| PET-OAM-C | **8.700** | 9.040 |
| PET-MPtrj-C | 12.072 | 16.959 |
| PET-OMAD | 15.610 | 17.633 |
| PET-OAM-NC | 23.041 | 23.756 |
| PET-MPtrj-NC | 41.932 | 20.462 |
| GRACE-2L-OAM | 9.138 | **7.459** |
| SevenNet-MF-ompa | 9.540 | 9.463 |
| Orb-v3 | 9.749 | 7.582 |
| MACE-MPA-0 | 10.270 | 15.026 |
| DPA-3.1-3M | 10.766 | 10.131 |
| MatterSim-v1-5M | 12.751 | 14.948 |
| PET-MAD | 17.325 | 32.559 |
| DPA-2.4-7M | 17.759 | 16.456 |
| SevenNet-l3i5 | 19.421 | 9.934 |
| MACE-MP-0 | 26.195 | 11.006 |
| Orb-v2 | 66.074 | 44.082 |

Table 9: Performance of different universal models on the Wiggle150 test from the LAMBench benchmark. The mean absolute error (MAE) and root mean squared error (RMSE) in predicting the energies of the conformers are given in kcal/mol. The best results per column are shown in bold, and the second-best are underlined.

| Model | MAE | RMSE |
|---|---|---|
| PET-OAM-NC | 6.127 | 7.741 |
| PET-OAM-C | 7.219 | 8.166 |
| PET-MP-C | 8.385 | 9.653 |
| PET-OMAD | 10.967 | 11.558 |
| PET-MP-NC | 11.085 | 13.400 |
| DPA-3.1-3M | **5.669** | **6.523** |
| Orb-v2 | 6.463 | 8.270 |
| PET-MAD | 8.798 | 9.818 |
| MatterSim-v1-5M | 10.730 | 12.450 |
| SevenNet-MF-ompa | 10.970 | 12.800 |
| Orb-v3 | 11.922 | 12.894 |
| GRACE-2L-OAM | 12.140 | 13.994 |
| SevenNet-l3i5 | 13.881 | 15.098 |
| DPA-2.4-7M | 14.843 | 15.698 |
| MACE-MPA-0 | 14.915 | 16.677 |
| MACE-MP-0 | 26.597 | 28.513 |

various parts of the LAMBench indicate that PET-OAM models can still afford good predictions on molecular structures.

# E  MADBENCH EVALUATION

MADBench is another minimalistic benchmark that is used to assess the accuracy of universal atomistic models across various domains, ensuring internal consistency of the reference ab initio theory (Mazitov et al., 2025a). It contains subsets sampled from popular datasets for atomistic machine learning, covering inorganic materials (represented by subsets of the MAD (Mazitov et al., 2025b), MPtrj (Deng et al., 2023), matbench-discovery (Riebesell et al., 2023), and Alexandria (Schmidt

Table 10: Evaluation of various universal MLIPs on the MADBench benchmark from Mazitov et al. (2025a). Performance of the PET-OAM, PET-MPtrj and PET-OMAD models is computed in this work, and other results are reproduced from the reference paper (Mazitov et al., 2025a). For each subset and model, mean absolute errors are reported for raw energy (E) and force (F) predictions in meV/atom and meV/Å, respectively. We do not show force errors for the matbench-discovery subset, since forces are not available in the reference data.

| Model | MAD | | MPtrj | | Matbench | Alexandria | | OC2020 | | SPICE | | MD22 | |
|---|---|---|---|---|---|---|---|---|---|---|---|---|---|
| | E | F | E | F | E | E | F | E | F | E | F | E | F |
| PET-OMAD | **6.0** | **29.2** | 12.1 | 36.0 | **10.6** | 26.3 | 30.4 | **8.0** | **51.6** | **2.4** | **44.6** | 3.4 | **53.4** |
| PET-OAM-C | 43.1 | 58.3 | 10.1 | 13.5 | 37.1 | 12.6 | 21.3 | 25.8 | 67.7 | 13.2 | 53.6 | 10.9 | 59.3 |
| PET-OAM-NC | 46.3 | 71.4 | 4.0 | 11.9 | 38.6 | **11.4** | 12.5 | 20.0 | 76.5 | 13.7 | 75.6 | 13.4 | 93.0 |
| PET-MPtrj-C | 55.8 | 101.3 | 5.5 | 9.1 | 41.8 | 36.5 | 51.7 | 42.4 | 117.3 | 13.0 | 81.8 | 7.4 | 101.0 |
| PET-MPtrj-NC | 91.3 | 102.2 | **2.9** | **7.8** | 48.7 | 44.6 | 47.7 | 34.7 | 119.1 | 34.7 | 103.1 | 19.4 | 122.8 |
| PET-MAD | 17.6 | 65.0 | 22.3 | 77.6 | 31.3 | 49.0 | 65.3 | 18.3 | 114.5 | 3.7 | 59.4 | **1.9** | 65.6 |
| MACE-MP-0-L | 81.6 | 181.5 | 15.1 | 50.8 | 58.5 | 65.4 | 79.5 | 82.4 | 169.6 | 10.6 | 166.8 | 9.4 | 182.9 |
| MatterSim-5M | 47.3 | 133.7 | 21.3 | 61.4 | 38.2 | 21.2 | 39.9 | 31.5 | 119.2 | 21.3 | 145.6 | 28.6 | 160.4 |
| Orb-v2 | 52.9 | 96.2 | 5.6 | 21.9 | 37.9 | 13.2 | **10.5** | 19.8 | 99.3 | 59.0 | 140.8 | 174.3 | 220.7 |
| SevenNet-l3i5 | 82.1 | 173.5 | 9.8 | 25.5 | 47.5 | 47.6 | 70.3 | 45.7 | 162.7 | 11.3 | 139.1 | 11.1 | 146.2 |

et al., 2024) datasets), molecules (represented by subsets of the SPICE (Eastman et al., 2023) and MD22 (Chmiela et al., 2023) datasets), and catalytic applications (a subset of the Open Catalyst 2020 (Chanussot et al., 2021) dataset), recomputed with a unified set of DFT settings to obtain internally coherent data on target energies, forces and stresses.

We evaluate the performance of our proposed models on MADBench and compare it against other model results from Mazitov et al. (2025a). The results are presented in Table 10. In almost all cases, the PET-OMAD model demonstrates the best overall accuracy in predicting both energies and forces, while falling behind PET-MPtrj and PET-OAM only on the MPtrj and Alexandria subsets, which have considerable overlap with the noted subsets in the training data. A similar effect is observed in the Orb-v2 results for Alexandria. The best accuracy in predicting forces from this subset is likely due to overfitting on the Alexandria data, which is part of this model's training set. Upon comparing the PET-OMAD results against PET-MAD, one can see the combined effect of pre-training on the OMat24 dataset and using the updated architecture: PET-MAD errors are cut almost by half on the majority of the subsets.

## F  SPICE EVALUATION

In order to obtain timings for the MACE, eSEN and PET architectures, we perform inference with the same set-up described in Fu et al. (2025), using the same hardware (a single Nvidia A100 80 GB GPU). The timings we present in this work are evaluated by us for PET and MACE, while the eSEN timings are taken from Fu et al. (2025). In Fig. 5 we further include the MACE-L timings reported in Fu et al. (2025), showing that any potential differences due to the evaluation setup do not affect the results significantly (we suspect them to originate from different power GPU power settings on different clusters). Similarly, although we make use of torchscript compilation for PET evaluations (while MACE and eSEN do not), the differences we observed compared to eager-mode Pytorch (Paszke et al., 2019) evaluations are minimal.

For completeness, we report accuracies of all evaluated models in Tab. 11. MACE and eSEN accuracies are taken from Kovács et al. (2025) and Fu et al. (2025), respectively.

## G  DETAILS ON GEOMETRY OPTIMIZATION AND PHONON CALCULATIONS

Geometry optimizations are performed with ASE (Larsen et al., 2017) version 3.26.0 using metatomic (Bigi et al., 2025) calculators. Constrained optimizations use the FixSymmetry function and freezing the off-diagonal strain degrees of freedom in FrechetCellFilter. The convergence threshold for forces and (appropriately scaled) stress components is set to $10^{-5}$ eV/Å.

Table 11: Test set accuracies for models trained on the dataset presented in Kovács et al. (2025). Energy MAEs are shown in units of meV per atom; force MAEs are shown in units of meV/Å. For each metric, the best model is highlighted in bold and the second best is underlined.

| Subset | MACE-S | | MACE-L | | eSEN-S | | eSEN-L | | PET-S | | PET-L | |
|---|---|---|---|---|---|---|---|---|---|---|---|---|
| | E | F | E | F | E | F | E | F | E | F | E | F |
| PubChem | 1.41 | 35.68 | 0.88 | 14.75 | 0.22 | 6.10 | 0.15 | 4.21 | 0.19 | 6.79 | **0.09** | **3.53** |
| DES370K Monomers | 1.04 | 17.63 | 0.59 | 6.58 | 0.17 | 1.85 | 0.13 | 1.24 | 0.19 | 2.23 | **0.10** | **1.00** |
| DES370K Dimers | 0.98 | 16.31 | 0.54 | 6.62 | 0.20 | 2.77 | 0.15 | 2.12 | 0.18 | 2.17 | **0.12** | **1.20** |
| Dipeptides | 0.84 | 25.07 | 0.42 | 10.19 | 0.10 | 3.04 | 0.07 | 2.00 | 0.11 | 3.58 | **0.05** | **1.55** |
| Solvated Amino Acids | 1.60 | 38.56 | 0.98 | 19.43 | 0.30 | 5.76 | 0.25 | **3.68** | 0.36 | 8.78 | **0.17** | 4.37 |
| Water | 1.67 | 28.53 | 0.83 | 13.57 | 0.24 | 3.88 | 0.15 | **2.50** | 0.27 | 6.11 | **0.13** | 3.05 |
| QMugs | 1.03 | 41.45 | 0.45 | 16.93 | 0.16 | 5.70 | 0.12 | 3.78 | 0.15 | 6.71 | **0.08** | **2.91** |

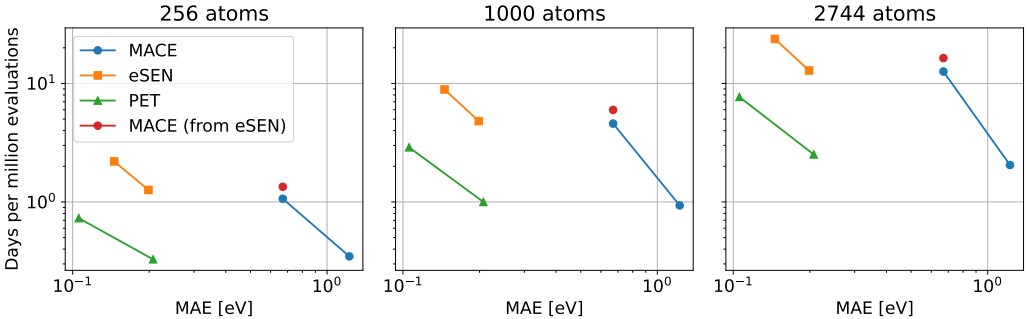

Figure 5: Accuracy-speed Pareto front for models trained on the SPICE dataset, at varying number of atoms. Timings corresponding to MACE evaluations in Fu et al. (2025) do not differ significantly from those presented in this work and might stem from different hardware power settings.

Phonon bands are computed with ASE's phonons module. The displacement used for finite-difference force constants is 0.03 Å for conservative models and 0.1 Å for non conservative ones. The larger value used for non conservative models is due to the rougher potential energy surface of NC models. This is evident in Figure 6a, where the NC model do not exhibit the expected linear force-displacement relation near equilibrium, unlike conservative models. Interestingly, the NC

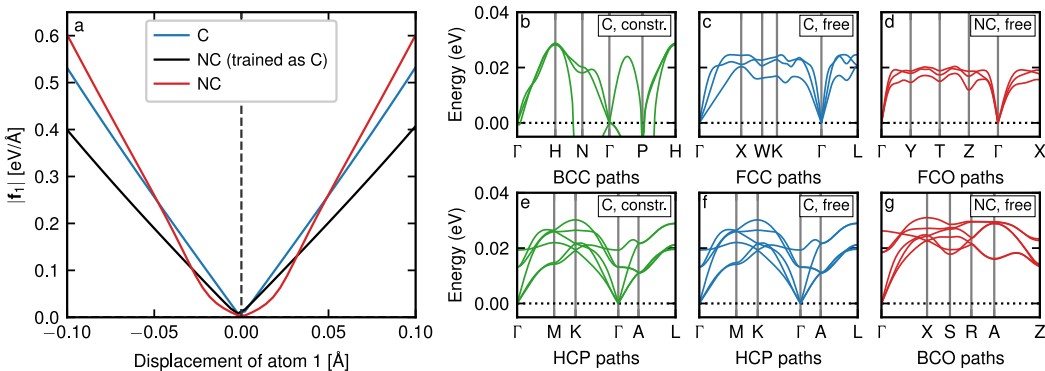

Figure 6: (a) Magnitude of the force on one HCP Ti atom as a function of its displacement from equilibrium. (b-g) Phonon bands plotted along high-symmetry lines in the BZ corresponding to the relaxed structures.

heads of a conservatively trained model instead still recover linear behavior, suggesting that conservative training can benefit direct-force models.

In Figure 6(b–g) we plot again the phonon spectra from Figure 2, this time along the high-symmetry lines of the BZ corresponding to the Bravais lattice of each relaxed structures. The lattices are identified using spglib (Togo et al., 2024) with a symmetry tolerance ("symprec") of 0.01. With this setting, unconstrained relaxations using the C model return FCC and HCP structures, while NC relaxations yield face-centered orthorombic (FCO) when starting from BCC and base-centered orthorombic (BCO) when starting from HCP. Relaxing the tolerance to 0.1 assigns the former as body-centered tetragonal and the latter as HCP, whereas tightening it to 0.001 reduces all unconstrained relaxations to triclinic $P\bar{1}$.

