# OpenReview forum: "Pushing the limits of unconstrained machine-learned interatomic potentials"
_ICLR.cc/2026/Conference — ICLR 2026 Conference Withdrawn Submission_

### Official Review · Reviewer_3Py5 · 2025-10-15

**Soundness:** 3
**Presentation:** 3
**Contribution:** 3
**Rating:** 2
**Confidence:** 5

**Summary:**

This paper investigates the performance of unconstrained machine-learned interatomic potentials (MLIPs) — models that do not explicitly enforce rotational equivariance or energy conservation. The authors scale up a PET-based GNN architecture to hundreds of millions of parameters and evaluate it on large-scale materials (OMat24, MPtrj) and molecular (SPICE) datasets. The results show that unconstrained models can reach comparable accuracy to equivariant ones while offering 2–3× faster inference. The paper also discusses inference-time corrections (rotational averaging, Hessian symmetrization) to recover physical observables.

**Strengths:**

This paper is a solid engineering and benchmarking effort but provides limited conceptual novelty.

**Weaknesses:**

The discussion on conservative vs. non-conservative forces and equivariant vs. non-equivariant architectures has already been extensively studied in recent works.

The idea of combining direct-force pretraining with conservative fine-tuning was already proposed and empirically validated in eSCN rameworks. Therefore, the conceptual novelty of this paper is limited.

The proposed PET modifications (RMSNorm, SwiGLU, increased node features) are incremental architectural tweaks rather than fundamentally new ideas. The core PET design and inference-time averaging have been previously explored.

**Questions:**

See Section Weakness.

---

### Official Review · Reviewer_no2Y · 2025-10-28

**Soundness:** 3
**Presentation:** 3
**Contribution:** 2
**Rating:** 2
**Confidence:** 4

**Summary:**

The authors train a non-rotationally equivariant MLIP on relevant chemical datasets. They show competitive performance compared to equivariant architectures and discuss the role of different constraints in performance and inference speed.

**Strengths:**

- Training efficient MLIPs is of interest to the community
- The discussion surrounding the amount of augmentation and what can be expected with / without constraints is interesting.
- The paper is well written and clearly motivated

**Weaknesses:**

The way the paper is currently written, it is difficult to pinpoint the exact contribution that the authors are advocating for. There are a number of existing works that already show strong performance with non-equivariant architectures [1,2,3,4]. It is already known that these non-rotationally equivariant models can perform quite well. I find the discussion in Section 3.1 surrounding how hard different symmetries are to learn, and in fact I think focusing on this discussion with clear experiments would make the paper stronger and more novel. However, as it stands, I have some worries about the relevance of the current experiments:

- While the performance on materials is competitive, non-equivariant architectures have already been shown to perform extremely well on these benchmarks [1,2,3].
- Again the performance on SPICE is competitive, but models are already near perfect and there are known issues with the dataset [6], making differences between such accurate models hard to evaluate. It also seems like training resources were not clearly controlled between baselines (I suggest the authors report the speed of a few eSEN / EScAIP / MACE training steps on the same hardware used to train their models to make a clearer comparison). Something like OMol is likely a much better place to evaluate performance on molecules [5].

[1] Eric Qu, & Aditi S. Krishnapriyan. (2024). The Importance of Being Scalable: Improving the Speed and Accuracy of Neural Network Interatomic Potentials Across Chemical Domains.

[2] Mark Neumann, James Gin, Benjamin Rhodes, Steven Bennett, Zhiyi Li, Hitarth Choubisa, Arthur Hussey, & Jonathan Godwin. (2024). Orb: A Fast, Scalable Neural Network Potential.

[3] Benjamin Rhodes, Sander Vandenhaute, Vaidotas Šimkus, James Gin, Jonathan Godwin, Tim Duignan, & Mark Neumann. (2025). Orb-v3: atomistic simulation at scale.

[4] Max Eissler, Tim Korjakow, Stefan Ganscha, Oliver T. Unke, Klaus-Robert Müller, & Stefan Gugler. (2025). How simple can you go? An off-the-shelf transformer approach to molecular dynamics.

[5] Daniel S. Levine, Muhammed Shuaibi, Evan Walter Clark Spotte-Smith, Michael G. Taylor, Muhammad R. Hasyim, Kyle Michel, Ilyes Batatia, Gábor Csányi, Misko Dzamba, Peter Eastman, Nathan C. Frey, Xiang Fu, Vahe Gharakhanyan, Aditi S. Krishnapriyan, Joshua A. Rackers, Sanjeev Raja, Ammar Rizvi, Andrew S. Rosen, Zachary Ulissi, Santiago Vargas, C. Lawrence Zitnick, Samuel M. Blau, & Brandon M. Wood. (2025). The Open Molecules 2025 (OMol25) Dataset, Evaluations, and Models.

[6] Domantas Kuryla, Fabian Berger, Gábor Csányi, & Angelos Michaelides. (2025). How Accurate Are DFT Forces? Unexpectedly Large Uncertainties in Molecular Datasets.

**Questions:**

Smaller comments / questions:
- Line 190: Where does the 0.1 Å come from? This seems very dependent on the downstream application.
- Line 48: Something like Gemenet still has many inductive biases and "constraints"
- What hardware was used to run the speed benchmarks in Fig. 3? Is this the single A100 80 GB card mentioned in Appendix F? For a speed benchmarks, theses details are useful to have in the main text for the reader.

---

### Official Review · Reviewer_LXtX · 2025-11-01

**Soundness:** 3
**Presentation:** 3
**Contribution:** 1
**Rating:** 2
**Confidence:** 4

**Summary:**

This paper investigates how much inductive-bias is required in machine learned interatomic potentials. These inductive-biases include: rotations, translations, permutations, inversions, and energy conservation. In the paper, an unconstrained model (based on the point edge transformer) is trained on the MPtrj and OAM materials datasets. The model is evaluated on a number of tasks including geometry optimizations, phonons, and the Matbench Discovery leaderboard. Both the PET-MPtrj and PET-OAM models are highly competitive on the Matbench Discovery leaderboard. Similarly, a PET model is trained on SPICE, a dataset for molecules, where it achieves SOTA result against models with similar inference speeds. The paper concludes that rotational equivariance can be learned whereas non-conserving models look good on test set metrics but fail in other settings.

**Strengths:**

- The work focuses on an important topic.
- The paper does an excellent job laying out the background information.
- The paper is technically solid.

**Weaknesses:**

- There is not sufficient novelty presented in the paper. The model architecture is not new; it is based on the point edge transformer, which was published previously. The main conclusions that rotational equivariance can be learned and that energy conservation needs to be included (this can be made more efficient with non-conservative pre-training) has been reported by a number of others (Orb-v3: atomistic simulation at scale, The dark side of the forces: assessing non-conservative force models for atomistic machine learning, The Importance of Being Scalable:Improving the Speed and Accuracy of Neural Network Interatomic Potentials Across Chemical Domains, and Learning Smooth and Expressive Interatomic Potentials for Physical Property Prediction). While the empirical results on Matbench Discovery and SPICE are promising and demonstrate clear value, additional contributions would strengthen the case for publication.
- The paper does not include any molecular dynamics (MD) evaluations. It would be nice to include NVE MD energy drift experiments similar to those in Learning Smooth and Expressive Interatomic Potentials for Physical Property Prediction to see how the unconstrained PET model behaves.

**Questions:**

1. Can you explain the constrained relaxation procedure in more detail?
2. In Figure 2, is there a DFT reference available?
3. How much does rotational or O(3) averaging dig into the speed benefits of an unconstrained model?

---

### Official Review · Reviewer_xQpZ · 2025-11-09

**Soundness:** 2
**Presentation:** 2
**Contribution:** 2
**Rating:** 2
**Confidence:** 4

**Summary:**

This paper looks at an architecture that is a modification of the Point Edge Transformer (PET). It explores this architecture in the context of not building in rotational equivariance or energy conservation. The model is assessed on materials benchmarks such as the ones in Matbench Discovery, as well as on the SPICE molecular dataset.

**Strengths:**

- The analysis of the phonon calculations is interesting, and provides a nice analysis of what the ML community should be looking at

**Weaknesses:**

- The primary thing that this paper seems to look at is having models that are unconstrained with respect to rotational equivariance and energy conservation, not unconstrained in all scenarios, so the title and the way the model is referred to is already misleading. There are a number of works that have also studied and shown this, and it’s not clear what is new here: the title of this paper should be renamed to reflect these very specific architectural removals (such as rotational equivariance), as it doesn’t seem like it is pushing the limits of a truly unconstrained architecture. On this note, it does not seem like it is “pushing the limits” of an architecture without these standard constraints either.

- This paper does not mention the Open Molecules dataset, which has been out for a number of months before the ICLR deadline, and represents a thorough dataset and benchmark suite. There are lines in the paper saying “applications of MLIPs to molecular systems have not yet seen the development of established benchmark suites” and this is inaccurate. For any new model paper, training and testing on Open Molecules should become standard. It has both energy and force evaluation metrics but also a number of additional evaluations.

- Combining the above points, this seems like an architecture that is only marginally different from before and is not actually a top performing model on any noticeable datasets.

**Questions:**

- The way the architecture is referred to is confusing. The model is called a “Point Edge Transformer” but seems to essentially still be a GNN, as it has message passing and a radius cutoff. There is a line in the paper that says it is “a GNN whose successive layers process the edges through a standard Transformer” but this is unclear: looking at the architecture, it seems like it would be a GNN that uses the attention mechanism. Can you please clarify, and/or update the text to reflect the correct notation?

- What would results be on the Open Molecules 2025 dataset?

---

### Note · Authors · 2025-11-29

**Comment:**

Dear reviewers and AC,

We have carefully examined the reviews and we agree with the overall assessment that our contribution requires further development. We have therefore decided to withdraw our paper from consideration at ICLR. We would like to thank the reviewers for their time and their helpful comments, which will guide our future revisions.

**Withdrawal Confirmation:**

I have read and agree with the venue's withdrawal policy on behalf of myself and my co-authors.